# Oxidative stress, dysfunctional energy metabolism, and destabilizing neurotransmitters altered the cerebral metabolic profile in a rat model of simulated heliox saturation diving to 4.0 MPa

**Xia Liu[1], Yiqun Fang[1]\*, Jiajun Xu[1], Tao Yang[1], Ji Xu[1], Jia He[1], Wenwu Liu[1], Xuhua Yu[1], Yukun Wen[1], Naixia Zhang[2], Ci Li[1]\***

**1** Department of Diving and Hyperbaric Medicine, Navy Medical Center of PLA, Naval Medical University (Second Military Medical University), Shanghai, China, **2** Department of Analytical Chemistry, Shanghai Institute of Materia Medica, Chinese Academy of Sciences, Shanghai, China

\* 1287225836@qq.com (YF); 1574006873@qq.com (CL)

**Data Availability Statement:** Metabolomics data have been deposited to the EMBL-EBI MetaboLights database (DOI: 10.1093/nar/

## Abstract

The main objective of the present study was to determine metabolic profile changes in the brains of rats after simulated heliox saturated diving (HSD) to 400 meters of sea water compared to the blank controls. Alterations in the polar metabolome in the rat brain due to HSD were investigated in cortex, hippocampus, and striatum tissue samples by applying an NMR-based metabolomic approach coupled with biochemical detection in the cortex. The reduction in glutathione and taurine levels may hypothetically boost antioxidant defenses during saturation diving, which was also proven by the increased malondialdehyde level, the decreased superoxide dismutase, and the decreased glutathione peroxidase in the cortex. The concomitant decrease in aerobic metabolic pathways and anaerobic metabolic pathways comprised downregulated energy metabolism, which was also proven by the biochemical quantification of the metabolic enzymes Na-K ATPase and LDH in cerebral cortex tissue. The significant metabolic abnormalities of amino acid neurotransmitters, such as GABA, glycine, and aspartate, decreased aromatic amino acids, including tyrosine and phenylalanine, both of which are involved in the metabolism of dopamine and noradrenaline, which are downregulated in the cortex. Particularly, a decline in the level of N-acetyl aspartate is associated with neuronal damage. In summary, hyperbaric decompression of a 400 msw HSD affected the brain metabolome in a rat model, potentially including a broad range of disturbing amino acid homeostasis, metabolites related to oxidative stress and energy metabolism, and destabilizing neurotransmitter components. These disturbances may contribute to the neurochemical and neurological phenotypes of HSD.

gkz1019, PMID:31691833) with the identifier
MTBLS7185. The complete dataset can be
accessed here: https://doi.org/10.1093/nar/
gkz1019.

**Competing interests:** NO authors have competing
interests.

## Introduction

High pressure above 1.3 MPa (at approximately 120-meter seawater) is induced in humans
and mammals at risk of central nervous system (CNS) changes [1, 2]. The CNS might be one
of the most sensitive targets in the occurrence of excessive atmospheric pressure, gas bubbles
in the body, and decompression sickness (DCS) caused by deep-sea diving. In such conditions,
a series of psychomotor and cognitive manifestations are highly complex, with distal and prox-
imal tremors, electroencephalographic abnormalities, fasciculations, myoclonus, sleep disor-
ders, nausea, headache, dizziness, and reduced performance on cognitive tests [2–4]. Moen
*et al.* measured regional neurological abnormalities by diffusion- and perfusion-weighted
magnetic resonance imaging (MRI). Perfusion deficits in cerebral microvascular function with
arterial microemboli [5] were found in *North Sea* divers, proven by reduced mean transition
time due to reduced complexity of the microvascular or capillary system. Alvhild Alette
Bjørkum *et al.* found disturbing protein homeostasis, e.g., in synaptic vesicles, and destabiliz-
ing cytoskeletal components after heliox saturation diving in a rat model. However, Arvid
Hope *et al.* reported that no visible CNS injuries of morphological changes under MRI scan
were observed in rats with massive neurological symptoms of decompression sickness follow-
ing heliox saturation decompression [6]. Thus, we hypothesize that potential molecular profile
alterations behind such significant but ambiguous physiological abnormalities after a heliox-
saturation dive might be observed in the CNS tissue.

The identification of biomarkers for monitoring the status of cellular physiological mecha-
nisms is important for understanding biochemical events. It is always a challenge considering
the complexity and diversity of molecular pathways involved in the response of biological sys-
tems to diverse factors in a specific moment or condition. Previous attempts linking individual
biomarkers with functional CNS perturbation have provided some insight into oxidative dam-
age and energy metabolism postsaturation dive [3, 4, 7]. Similarly, several studies have com-
pared the amino acid neurotransmitter profiles of rats with high-pressure neurological
syndrome to those of healthy individuals [2]. Illustrating the neurological metabolic finger-
print of CNS leisure might favor etiological hypotheses [8, 9], prevention [10], and therapeutic
approaches [11]. However, the integrated neurological metabolic perturbation induced by a
great depth of heliox saturation diving has not been investigated. Such information output
from complex biological systems can be rapidly recorded because of advances in technological
means [12]. The application of high-resolution nuclear magnetic resonance (NMR) spectros-
copy can provide extremely amounts of high complexity but interpretable and robust meta-
bolic profiling data. The spectral data of biofluids and tissue metabolite extracts can be
obtained by means of chemometric and bioinformatic methods to reveal physiological or path-
ological status information. NMR spectroscopy gives immediate qualitative and quantitative
information on approximately $10^2$ different small molecules present in a biological sample.
NMR detection enables a broad unbiased approach without a priori selection of specific bio-
chemical pathways. Additionally, NMR allows high-throughput analysis and high reproduc-
ibility, and it is an intrinsically quantitative technique over a wide dynamic range due to the
linear response of NMR signals within a concentration. This technology has been successfully
applied to neurological diseases such as cerebellar ataxia [13], Huntington's disease [14, 15],
and Alzheimer's disease [16, 17] in both preclinical and clinical studies.

Despite the promising applicability of NMR-based metabolomics, its application to evaluate
the central nervous metabolic perturbation effects of a heliox saturation diving has not been
reported previously. The aim of the present study was to investigate metabolomic changes in
different anatomic compartments of the brain (cortex, hippocampus, and striatum) and bio-
chemical index level changes in the cortex in a rat model after simulated 400 meters of sea

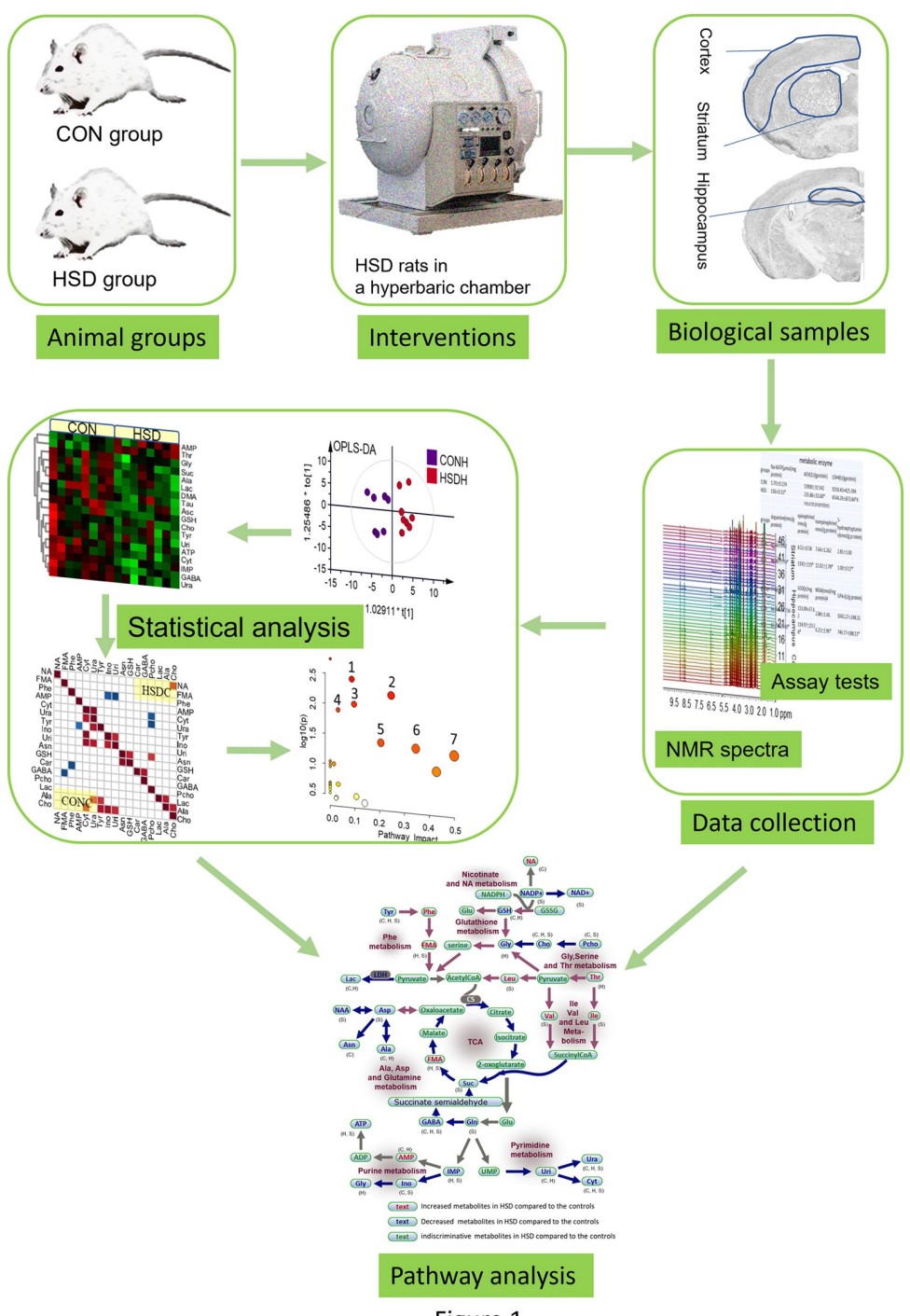

Figure 1.

**Fig 1. Schematic experimental design of the present research.**

water (msw) heliox saturation diving, as shown by the schematic experimental design of the present research in Fig 1. The illustration of the metabolic fingerprint of the target organ might provide a set of biomarkers available that could contribute to improvements in diving procedures.

## Experimental procedures

We confirm that we have read the Journal's position on issues involved in ethical publication and affirm that this report is consistent with those guidelines.

### Reagents and materials

Analytical-grade sodium chloride, DMSO, $NaN_3$, $NaH_2PO_4 \cdot 2H_2O$, and $Na_2HPO_4 \cdot 12H_2O$ were purchased from Sinopharm Chemical Reagent Co. Ltd. (Shanghai, China). HPLC-grade $CHCl_3$ and $CH_3OH$ were obtained from Merck (Darmstadt, Germany). $D_2O$ (99.9% in D) containing sodium 3-(trimethyl-silyl) propionate-2, 2, 3, 3, $d_4$ (TSP) as an internal standard for chemical shift reference was provided by Sigma–Aldrich (MO, USA). A buffer system containing 0.2 M $Na_2HPO_4/NaH_2PO_4$ in $D_2O$ at pH 7.4 was prepared to prevent the pH effect on the chemical shifts of metabolites at different concentrations. The assay kits for the determination of sodium-potassium ATPase (Na-K-ATPase), cholinesterase (AChE) and lactate dehydrogenase (LDH) were purchased from Abcam (USA). The assay kits for dopamine (DA) were purchased from RD, USA, the assay kits for epinephrine (E) and norepinephrine (NE) were from Abnova, Taiwan, 5-hydroxytryptamine (5HT) assay kits were from BioSource, and gamma-aminobutyric acid (GABA) assay kits were from Santa Cruz, USA. The assay kits of superoxide dismutas (SOD), malonyldialdehyde (MDA), and glutathione peroxidase (GPx) were purchased from Cayman, USA.

### Ethics approval

All experimental protocols were approved by the Animal Ethics Committee of Naval Medical Center of PLA, Naval Medical University (Approval no: SYXK(Shanghai)2017-0019, Approval year: 2020). All animals received good care according to the Guide of the Care and Use of Laboratory Animals of Naval Medical University. We confirm that all methods are reported in accordance with ARRIVE guidelines (https://arriveguidelines.org) for the reporting of animal experiments. The participants researched in this study were rats, not humans; thus, there was no consent to participate.

### Animals and grouping design

Adult male Sprague–Dawley rats with an average age of 8 weeks and weighing 200–220 g were included in this research. Animals were purchased from Shanghai Slac Laboratory Animal Co., Ltd. (Shanghai, China). Maximum efforts were carried out to minimize animal suffering and the number of animals necessary for the capture of reliable data. Standard rat chow and drinking water were available for all animals ad libitum. Animals with 3 rats per cage were housed in a specific pathogen-free (SPF) animal room (temperature, 22–24°C, humidity, 45–55%) and controlled conditions of light (12/12 hour light-dark cycle) for one week before the experiment. Prior to commencement of the diving experiments, rats were acclimatized for five days in the lab environment, including a hyperbaric chamber.

Sixteen rats were weighed, labeled, and restricted randomly assigned to a control (exposure to normobaric air but enduring similar light and noise, named the CON group, n = 8) group and an experimental (subjected to a simulated heliox saturation diving to 4.0 MPa (400 msw), named the HSD group, n = 8).

### Stimulated saturation diving of 400 msw

Rats in the HSD group were pressurized in helium-oxygen gas in a hyperbaric chamber. In brief, four phases of compression, storage, decompression, and bendwatch were included in

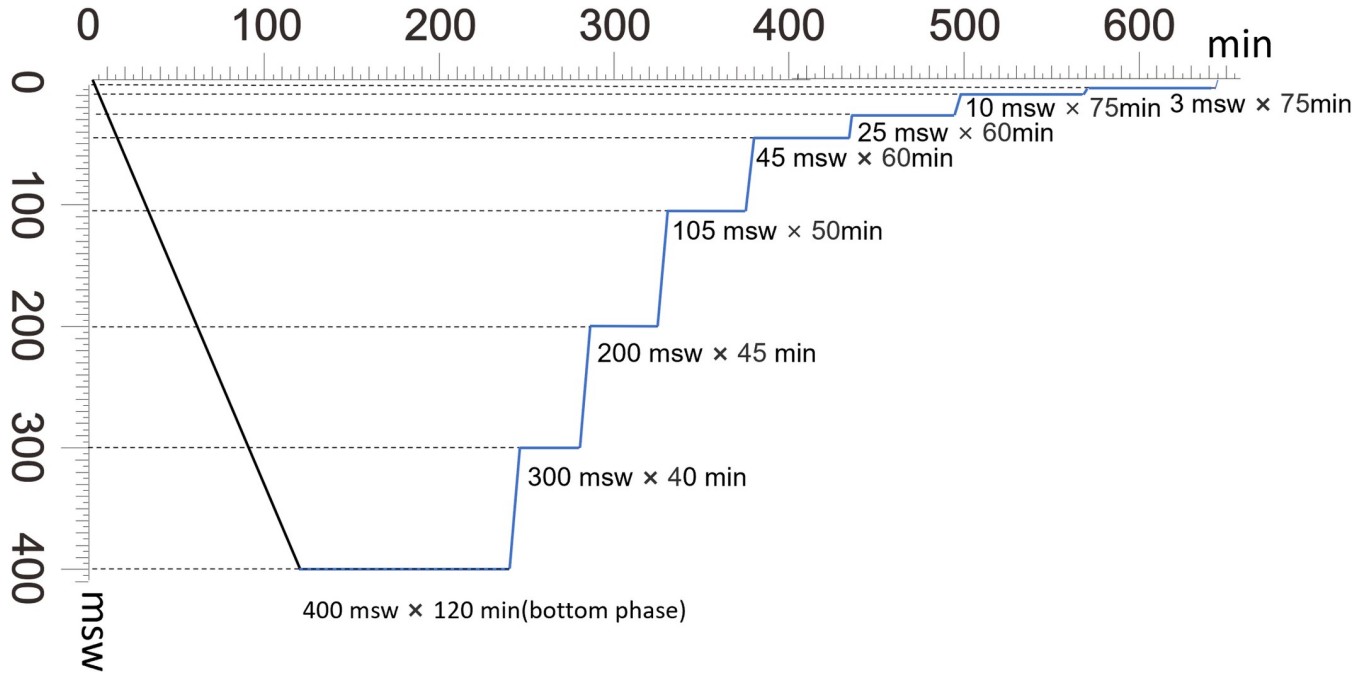

**Fig 2. Illustrative timeline overview of the hyperbaric decompression of stimulated heliox saturation diving to 400 msw.**

the saturation period. The rat's compression rate was 1 msw/min up to 10 msw with 20% He-O gas mixtures and 3.54 msw/min from 10 msw to 400 msw with pure He. The time compression from the surface to 400 msw took approximately 120 minutes. The bottom phase at 400 msw was 120 minutes. The oxygen concentration was supplemented with pure oxygen to maintain the oxygen partial pressure at 35~50 kPa during the compression process and the storage depth. The period of decompression bendwatch was 40 min at 300 msw, 45 min at 200 msw, 50 min at 105 msw, 60 min at 45 msw, 75 min at 10 msw, 75 min at 3 msw, and then at the surface. The ascent rate used in this decompression model is approximately 10 meters (33 ft) per min. the oxygen partial pressure was maintained at 38~67 kPa during the decompression stage. After decompression to 10 m, oxygen concentration was maintained at 20~24%. The saturation diving timeline overview is shown in Fig 2. Rats in the air control group (Group CON, named the control group, with eight rats) were bred within the atmospheric environment in the same experimental lab with the same chamber as the HSD groups. The control group did not receive any compressing or decompressing procedures but endured similar noise and light as the HSD group.

## Sample collection

After the decompression period, the rats were anesthetized with pentobarbital sodium (0.3%, 1.0 ml/kg rat weight) intraperitoneally, followed by removal of the brain. The left cerebral hemispheres were rapidly dissected. Cortex, hippocampus, and striatum tissues were dissected, and abbreviated respectively as HSDC, HSDH, HSDS, CONC, CONH and CONS in two animal groups. As the largest tissue among the three compartments, each cortex sample was then cut into two pieces with approximately equal weights (one piece of sample for metabolomics analysis, the other pieces of samples for biochemical assessments). All tissues were snap-frozen in liquid nitrogen and stored at −80°C until further analysis.

## Tissue extraction procedure

The extraction procedure of polar metabolites from tissue samples was a minor modification as reported [18]. Preweighed frozen tissue samples were thawed on ice and then subjected to mechanical homogenization in an icy-cold HPLC-grade methanol-chloroform-water solvent system (400 μL, 400 μL, and 285 μL, respectively, per 100 mg brain tissue) using a tissue homogenizer (Precellys 24, Bertin Technologies, Villeurbanne, France). The resultant homogenates were retained on ice for a period of 30 min and then centrifuged at 12,000 x g for a period of 10 min at 4˚C. The supernatant of each sample was then removed and lyophilized to obtain powder containing polar metabolites in a freeze dryer (FD-1A-80, BIOCOOL, China). Each powder sample was subsequently reconstituted with 550 μL PBS buffer containing 0.1% TSP, and all samples were then thoroughly rotamixed and centrifuged at 12,000 x g for a period of 20 min at 4˚C. An aliquot of 500 μL of supernatant was then transferred into a 5.0 mm-diameter NMR tube (Norrel, UK).

The extraction principle of protein in the cortex sample was as previously described with no modifications. The exact amount of cortex samples was poured into a homogenization buffer (HEPES 25 mmol/L, pH 7.4, $MgC1_2$ 5 mmo1/L, DTT 2 mmol/L, EDTA 1.3 mmol/L, EGTA 1 mmol/L, 0.1% Triton X-100, aprotinin, pepstatin A and leupeptin 10 μg/ml each) and homogenized manually in an ice bath. The mixture was centrifuged at 1,000 x g for a period of 10 min at 4˚C, and the supernatant was used for the quantified protein concentrations using the Bradford test.

## Biochemical assay

A sensitive, competitive enzyme-linked immunosorbent assay (ELISA) was applied with assay kits for the quantification of the metabolic enzymes Na-KATPase and AChE and the neurotransmitters DA, E, NE, 5HT, and GABA. The levels of SOD, MDA, and GPx in cortical tissue were also determined by ELISA with assay kits according to the manufacturer's instructions.

## NMR measurement

The efficient quantification of tissue metabolites was achieved using an analytical platform based on a liquid high-resolution Bruker Avance-III NMR spectrometer equipped with a high-sensitivity cryogenic probe operating at a frequency of 600.17 MHz for [1]H observation at 298 K. A water-suppressed one-dimensional [1]H ZGPR (TOPSPIN version 3.0, Bruker Biospin) pulse sequence (RD-90˚-ACQ) was applied to acquire NMR data for each sample. Four dummy scans and 128 transients were recorded into a time domain of 32 K data points using a spectral width of 20 ppm with a relaxation delay of 10.0 s and an acquisition time of 2.73 s. An exponential line-broadening function of 0.3 Hz and zero-filling to 64 K data points were applied to all the free induction decays (FIDs) prior to Fourier transformation. Additional two-dimensional NMR techniques with pulsed field gradient correlation spectroscopy (gCOSY) and 2D homonuclear total correlation spectroscopy (TOCSY) were employed using standard pulse programs on selected samples to confirm the chemical shift assignments. An automated sample changer for continuous sample delivery was used in all spectra acquisition.

## Metabolite identification and confirmation

NMR signals based on the location of individual resonances on the spectra were identified in Chenomx NMR Suite v. 8.4 software package (Evaluation version). The confirmations of some metabolite assignments were carried out considering chemical shifts, coupling constants and multiplicity patterns of metabolites as information on scalar couplings extracted from [1]H–[1]H

COSY, $^1$H–$^1$H TOCSY, public NMR databases such as COLMAR and Human Metabolome Database (HMDB), and the literature [18, 19].

## Multivariate statistical analysis

The preprocessing protocol used for preprocessing each 1D $^1$H NMR spectrum was the same as that described in our previous work [20]. The $^1$H NMR spectra were phased and baseline-corrected using MestReNova (Mestrelab Research, S.L., Spain), and the spectral region of each metabolite was integrated into one bucket. The resonances of the organic solvent signal and the residual H2O/HOD signal region were removed in all 1D $^1$H NMR spectra. A 0.003 Hz bucket procedure and normalization to the sum of the spectral intensity multiplied by 10000 were applied in all spectra. To avoid misinterpretation of the discriminant metabolites due to overlapping signals, only the largest bucket values in one peak were selected for the next-step analysis. Subsequently, integral buckets of 47 metabolites were extracted and subjected to uni-variate and multivariate data analysis. Indiscrimination analysis by principal component analysis (PCA) and discrimination analysis by partial least squares-discriminant analysis (PLS-DA) and orthogonal projections to latent structures discriminant analysis (OPLS-DA) were performed using the SIMCA-P+14.0 software package (Umetrics, Umeå, Sweden) scaled to unit variance data. The PCA and PLS-DA score plots were illustrated with the first and the second principal components (t[1], t[2]), while the OPLS-DA score plot used t[1] and the orthogonal component (to[1]). The parameters Q2 (cum), R2Y(cum), and R2X (cum) were calculated to test the robustness of the discrimination models against overfitting [21]. As previously described, a quality assessment value of Q2 $\geq$ 0.4 is considered a reliable model. The sevenfold cross validation strategy and permutation test 200 times with the first component were applied to guard against model overfitting and further validate the reliability and trustworthiness of the models. If the Q2 regression line had a negative intercept and Q2 values fitted in the leftmost point were greater than all the Q2 values of the right points in the permutated test, the established OPLS-DA model was robust [22]. The default 10-fold cross validation strategy was applied to guard against model overfitting. In general, a quality assessment statistic (Q2) $\geq$ 0.4 is considered a reliable model, as previously described.

The correlation coefficients (r) and VIP values extracted from the OPLS-DA models were used to identify metabolites that contributed significantly to the separation between the two groups. Additionally, the fold changes of metabolites between groups were calculated using a function. $I_A/I_B$, where $I_A$ and $I_B$ represent the mean values for the metabolite integral in the A and B groups, respectively. The heatmap and box charts were plotted to facilitate the understanding of the metabolite variations among groups.

## Univariate statistics of metabolite integrals

The bucket average of metabolites in each group is expressed as the mean ± standard deviation (S.D.). Univariate analysis was also carried out using ANOVA (analysis of variance). Statistically significant differences between groups were evaluated by an unpaired $t$ test of two tails after log conversion in GraphPad Prism V 8.4.3 software (Graph Pad Software Inc., San Diego, CA, USA). Statistically significant differences were defined by values of $p$ ($< 0.05$).

According to the three criteria, including an absolute value of r greater than 0.50, a value of VIP greater than 1.0, and a $p$ value less than 0.05, the bucket variables with one of three features will be selected as discriminatory variables. Because more than one bucket value was listed from the same metabolite, two or more discriminative variables arising from the same metabolite will be present in the discriminatory panel. Selection will be carried out based on VIP rankings, and only the variables from the same metabolites with the highest VIP values

were therefore selected as discriminatory metabolite level features, which will be included in the next-step analysis.

## System statistical metabolic correlation analysis and hierarchical cluster analysis

Pearson's correlation coefficient calculation and hierarchical cluster analysis were carried out based on the relative integrals of metabolites in R Studio (Version 1.4.1717) with small scripts. For each group pair, the correlation matrix was illustrated in a pixel map as described in the literature [21]. Briefly, for each variable in one group, correlations with $p$ values less than 0.05 were considered significant statistical correlations and were kept to construct the final correlation network to illustrate the latent relationships of metabolites within the group and the disturbed metabolic relationships between groups. The redder squares indicate a more significant positive correlation. The bluer squares indicate a more significant positive correlation. The white squares indicate the correlation with no significance. Analysis of the most relevant cerebral metabolic pathways and networks invoked as respondents to HSD was performed using the MetaboAnalyst 3.0 tool. For the pathway topology analysis performed, the *Rattus norvegicus* (rat) mammalian pathway library was selected. This approach was also utilized to provide estimates of pathway impact, false discovery rate (FDR), and $p$ values.

## Results

### Cortical biochemical indexes from the HSD and CON groups

To validate the metabolic alteration effects of HSD on the brain tissues, biochemical parameters, including the energy metabolism-related metabolic enzymatic activities of Na-KATP, AChE, and LDH, the neurotransmitters of DA, E, NE, 5HT, and GABA, and the oxidative stress-related proteins of SOD, MDA, and Gpx in the ipsilateral cortex tissues of HSD and CON rats were also determined (Table 1). Compared with the control group, the levels of DA, E, NE, and MDA were increased significantly, whereas the contents of Na-KATP, AChE, LDH, 5HT, GABA, SOD, and Gpx were decreased in the HSD model. The limit of detection of assay kits used in the present work were listed in the S3 Table in the file of supplementary materials.

**Table 1. The quantification of biochemical parameters measured in cortex tissue CON and HSD animals by ELISA with the assay kits.** (mean ± S.D.). * indicates statistical significance.

| Groups | Metabolic enzymes | | |
|---|---|---|---|
| | Na-KATP (µmol/mg protein) | AChE (U/g protein) | LDH4 (U/g protein) |
| CON | 1.70±0.224 | 539.00±33.542 | 9250.43+425.394 |
| HSD | 116±0.13* | 235.86±15.00* | 6544.29±873.84* |

| Groups | Neurotransmitters | | | | |
|---|---|---|---|---|---|
| | dopamine (DA, nmol/g protein) | Epinephrine (E, nmol/g protein) | Norepinephrine (NE, nmol/g protein) | 5-hydroxytryptamine (5HT, nmol/g p rotein) | GABA (nmol/g protein) |
| CON | 65.64±4.82 | 8.52±0.58 | 7.64±1.262 | 2.81±1.00 | 6.49±0.72 |
| HSD | 96.88±7.74* | 11.42±119* | 12.02±1.78* | 1.00±0.17* | 3.93±1.08* |

| Groups | Oxidative stress indicators | | | | |
|---|---|---|---|---|---|
| | SOD (U/mg protein) | MDA (nmol/mg protein) | GPX+(U/g protein) | SOD (U/mg protein) | MDA (nmol/mg protein) |
| CON | 153.09±37.62 | 2.88±1.44 | 1042.27±248.32 | 153.09+37.62 | 2.88±1.44 |
| HSD | 114.97±23.38* | 6.23±1.96* | 746.77+188.13* | 114.97±23.38* | 6.23±1.96* |

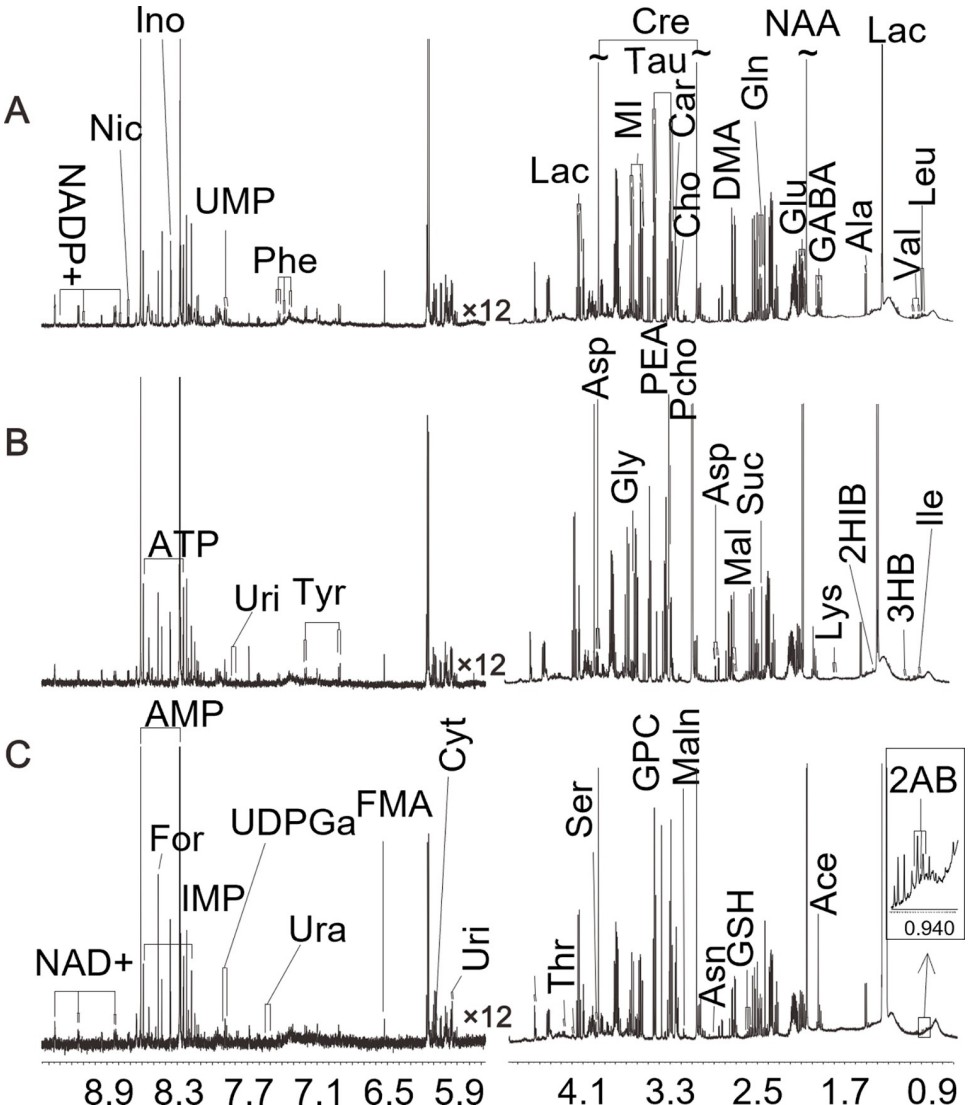

**Fig 3.** Representative 600 MHz 1H NMR spectra of the cortex (A), hippocampus (B), striatum (C) extracts obtained from the HSD rats, right part of 0.70–4.80 ppm spectral region and left part of 5.65–9.45 ppm spectral region (resonance intensity amplitude enhanced 12-fold of that of right part). The abbreviations of metabolites are shown in Additional file 1: S1 Table.

## Metabolites identified in [1]H NMR spectra of brain tissue samples

Fig 3 displays three representative [1]H NMR spectra from the cortex (Fig 3A), hippocampus (Fig 3B), and striatum (Fig 3C) of an HSD rat. A wide range of prominent metabolites were identified according to the [1]H NMR data of brain tissue samples, and they are organic acid anions (lactate (Lac), malonate (Maln), succinate (Suc), malate (Mal), fumarate (FMA), 2-hydroxybutyrate (2-HB), acetate (Ace), formate (For), taurine (Tau), ascorbate(Asc), 2-hydroxyisobutyrate (2HIB)), amino acids (leucine (Leu), isoleucine (Ile), valine (Val), alanine (Ala), glycine (Gly), tyrosine (Tyr), phenylalanine (Phe), aspartate (Asp), glutamine (Gln), glutamate (Glu), threonine (Thr), serine(Ser), lysine (Lys), asparagine (Asn)), neurotransmitters (γ-aminobutyrate (GABA)), energy-related metabolites (creatine (Cre), adenosine triphosphate (ATP), adenosine monophosphate (AMP)), phospholipid related metabolites (O-

phosphoethanolamine (PEA), O-phosphocholine (Pcho), sn-glycero-3-phosphocholine (GPC)), and others (myo-inositol (MI), nicotinamide adenine dinucleotide (NAD+), nicotinuric acid (Nic), N-acetylaspartate (NAA), nicotinamide-adenine-dinucleotide phosphate (NADP+), UDP-N- galactose (UDPGa), uridine (Uri), uracil (Ura), cystidine (Cyt), uridine 5'-monophosphate (UMP), inosine 5'-monophosphate (IMP), inosine (Ino), choline (Cho), carnitine (Car), glutathione (GSH). S1 Table shows detailed information on the metabolite assignments.

## Metabolic profile alterations revealed by metabolomic analysis

PCA, an exploratory and unbiased analysis approach of the $^1$H NMR spectra from all brain extracts across different brain regions, was first made to reveal the main metabolic trends driven by hyperbaric exposure to a 400 msw heliox saturation environment. A PC1 vs. PC2 scatting plot obtained from PCA (S1A–S1C Fig) of integral bucket data revealed a certain discrimination with some overlap between the two classifications. Supervised investigations of PLS-DA (S1A'–S1C' Fig) and OPLS-DA (Fig 4A–4C) models exhibited clear class discriminations of the metabolic profiles between groups. Fig 4 shows the score plots of the OPLS-DA model for the cortex region (A), the hippocampus region (B), and the striatum region (C), showing clear discrimination between the HSD groups and the controls. The high explained variation and the goodness of the prediction reflected by the values of R2X and Q2 (S1A'–S1C' Fig, Fig 4A–4C) and permutation test plots (Fig 4'-4C'A) indicated the robustness of the generated supervised models. The correlation coefficient (r) extracted from the S-line plots, variable importance in projection (VIP), and the *p* value from the nonparametric univariate tests were collected to assess the significant metabolites responsible for the class-discriminating patterns. Therefore, meeting any one of the three criteria (the absolute value of r greater than 0.50, the value of VIP greater than 1.0, together with *p* value less than 0.05), we selected a panel of statistically significant metabolites (Table 1) responsible for the separation between the CON and HSD groups. In Table 1, fold-change values greater than 1 indicate an increased level in the HSD group, while fold-change values less than 1 indicate a decreased level in the HSD group. The mean SD values of discriminative metabolites are listed in S2 Table. Using the hierarchical cluster analysis of metabolites and the average linkage method, the generated heatmap (Fig 5A–5C) with dendrograms allows for a better visualization of three brain region metabolic alterations caused by hyperbaric exposure in a helium oxygen-saturated environment.

## Metabolic disorders observed in different brain regions of HSD rats

The abovementioned multivariate analysis and univariate analysis using the bucket height of metabolites in the different groups provided a great work tube to identify discriminative metabolites revealing the potential neurologic metabolic alterations associated with HSD events. Elevated AMP, FMA, Nic, and Phe and a decrease in Ala, Asn, Car, Cho, Cyt, GABA, GSH, Ino, Lac, Pcho, Phe, Tyr, Ura, and Uri were found in the cortex tissue of the HSD group compared with the CON group (Fig 5A, Table 1). Meanwhile, in the hippocampus, Ala, GSH, Lac, Uri, Cyt, GABA, Tyr, and Ura also decreased and AMP increased in the HSD group, as they did in the cortex. Moreover, the upregulated Thr and downregulated Gly, ATP, Tau, Imp, Suc, Asc, and DMA were expressed in hippocampus extracts of the HSD group relative to the controls (Fig 5B, 5D, Table 1). Compared with the CONS group, the contents of Cyt, GABA, Ura, Cho, and Thr also decreased as they did in the cortex and hippocampus. Additionally, increased levels of branched-chain amino acids (BCAAs, including Leu, Ile, Val), and Lys and decreased levels of Gln, NAA, NAD+, NADP+, Asp, a series of metabolites of ATP, Tau, IMP, and Suc, which also decreased in the hippocampus, and another two metabolites, Ino and

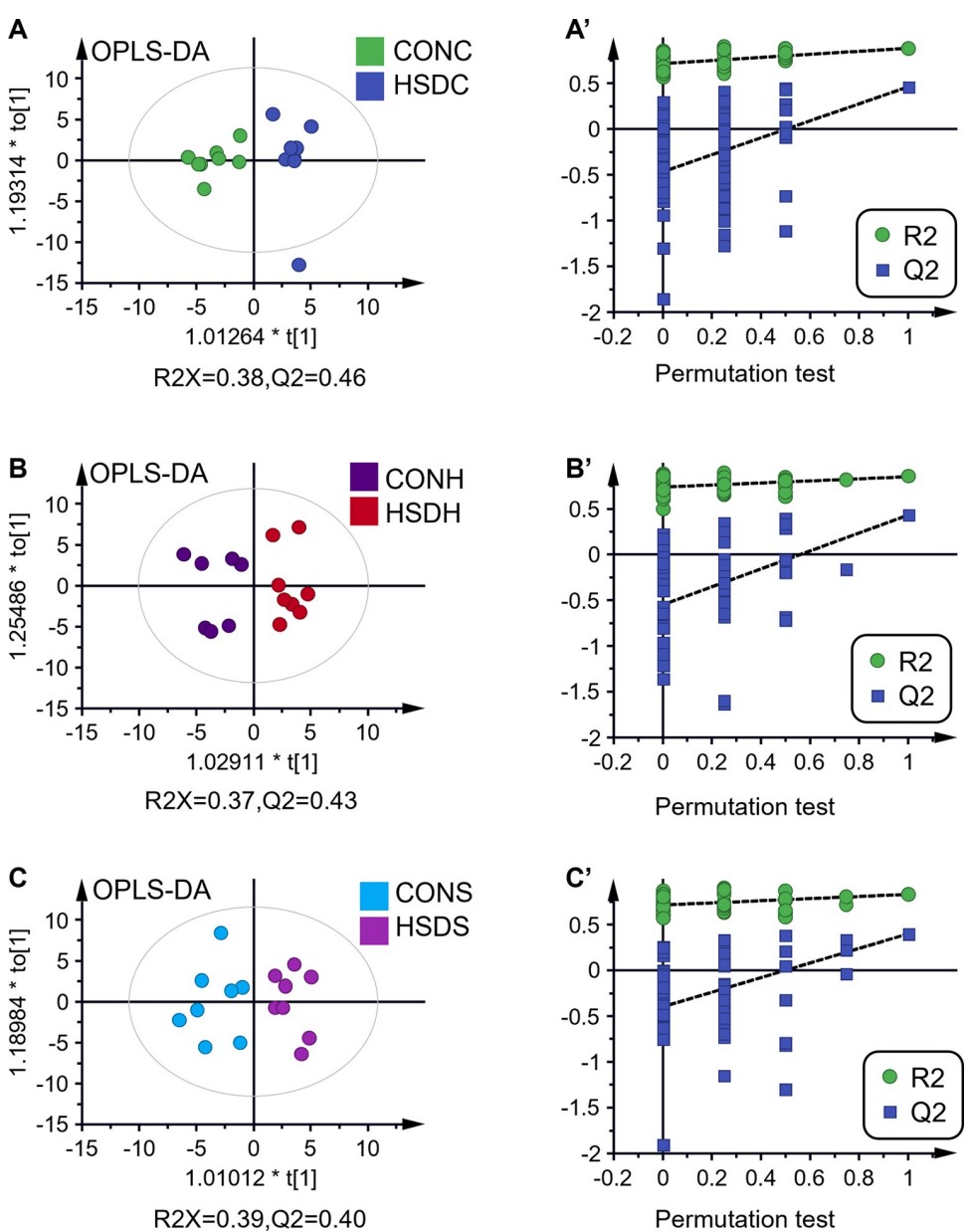

**Fig 4. Orthogonal partial least squares discriminant analysis (OPLS-DA) score plots and permutation test plots that discriminate the effect of 400 msw heliox-saturation exposure on 1H NMR spectra from control groups in the cortex (CONC, n = 8, HSDC, n = 8, A and A') (R2X = 0.38; Q2 = 0.46), hippocampus (CONH, n = 8, HSDH, n = 8, B and B') (R2X = 0.37; Q2 = 0.43), and striatum (CONS, n = 8, HSDS, n = 8, C and C') (R2X = 0.39; Q2 = 0.40).** The values of the Q2 parameter in the OPLS-DA score plots, which were equal to or greater than 0.40, coupled with the Q2 values of the leftmost point in the permutated models were greater than all the fitted Q2 values of the right points (permutation test 80 times), indicating that the established OPLS-DA models were valid.

Pcho, which also decreased in the cortex, were observed in the striatum tissues of HSD group rats (Fig 5C, 5D, Table 1). Such many metabolites always indicate implicated molecular pathways with complexity and diversity. The integral bucket values of discriminant metabolites were quantified, and statistically significant fold-changes of their concentrations between the heliox-saturation-hyperbaric-exposed and control groups are summarized in S2 Table and Table 2 for the three brain regions.

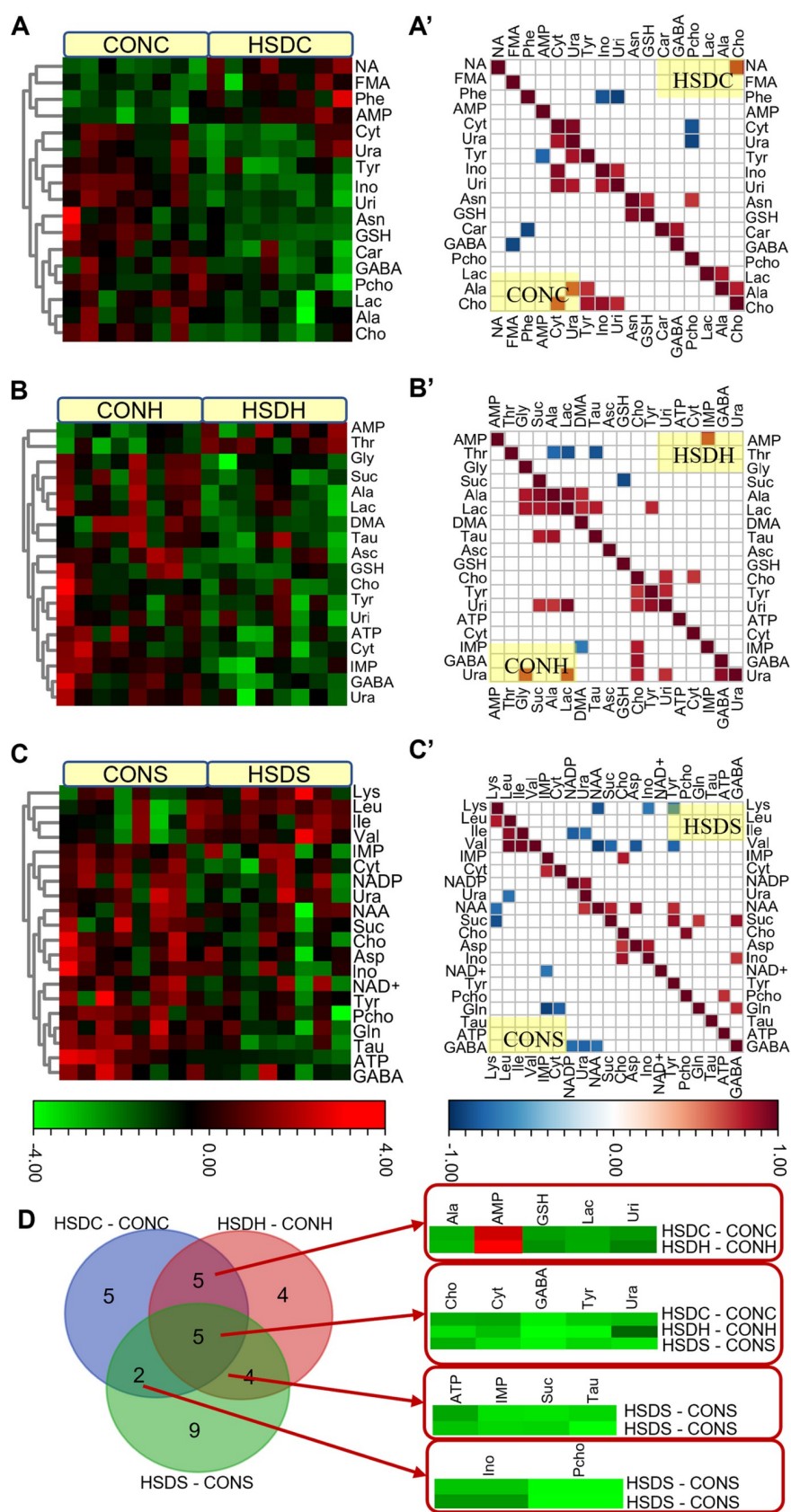

**Fig 5. Heatmap and statistical correlation plots derived from the bucket values of the discriminatory metabolites from cortex (A, A', HSDC samples lies in the upper panel and CONC lies in the lower panel), hippocampus (B, B', HSDH group lies in the upper panel and CONH lies in the lower panel), and striatum (C, C', HSDS samples lies in the upper panel and CONS lies in the lower panel) tissues of HSD and CON rats.** Metabolites on the heatmap are organized by hierarchical clustering based on the overall similarity in level patterns. Venn diagram (D) illustrating metabolite overlap among the HSDC-CONC, HSDH-CONH, and HSDS-CONS comparisons.

## Metabolite correlation analysis

The relationship between or among metabolites was so complex in Fig 5A'-5C'. For the energy metabolites, the positive correlations for Lac vs Ala in HSDC, AMP vs IMP in HSDH, ATP vs Pcho in HSDS, Suc vs GABA/Gln/Tyr/NAA in HSDS, and the negative correlations for Lac vs Thr and GSH vs Suc in HSDH, Suc vs Val in HSDS were present in the metabolite correlation plots. For the neurotransmitters, the positive correlations for GABA vs Car in HSDC, GABA vs Cho in CONH, and GABA vs Suc/Ino/Gln in HSDS were present in the metabolite correlation plots. The negative correlations for GABA vs FMA in the CONC and GABA vs NAA/Ura/NADP+ in the CONS were present in the metabolite correlation plots. For the metabolites related to oxidative stress, the positive correlations for GSH vs Asn in the HSDC and the CONC, Tau vs Lac in the HSDH, Tau vs Suc/Ala in the CONH, the negative correlations for GSH vs Suc in the HSDH, Tau vs Thr in the HSDH were present in the metabolite correlation plots.

## Metabolomics pathway analysis

Quantitative pathway analysis consisting of pathway enrichment analysis and pathway impact from pathway topology revealed highly statistically significant HSD-induced modulations to a series of metabolic pathways. Pathway impact scores, together with false discovery rate (FDR) and $p$ values, are described in Fig 6. Pathways were considered significantly enriched if $p$ values were lower than 0.05; the profiled metabolites (hits) relative to the total metabolites of the pathway (match status) were higher than 1; and the impact scores (indicating the impact of significantly affected metabolites in the pathway based on network topology measure of relative betweenness centrality) were higher than 0. The pathways in the cortex (Fig 6A) with the greatest metabolic impact value were phenylalanine, tyrosine, and tryptophan biosynthesis > phenylalanine metabolism > pyrimidine metabolism > alanine, aspartate, and glutamate metabolism. The pathways in the hippocampus (Fig 6B) with the greatest metabolic impact were phenylalanine, tyrosine, and tryptophan biosynthesis > glutathione metabolism > glycine, serine, and threonine metabolism > purine metabolism > pyrimidine metabolism > alanine, aspartate, and glutamate metabolism > butanoate metabolism. The pathways in the striatum (Fig 6C) with the greatest metabolic impact were alanine, aspartate, and glutamate metabolism > nicotinate and nicotinamide metabolism > purine metabolism.

## Discussion

This work aims to investigate the role of oxidative stress, energy metabolism and neurotransmitter profiles in the molecular mechanism of cerebral region-dependent metabolomics profile changes induced by heliox-saturation hyperbaric exposure. Several metabolites involved in energy metabolism, oxidative stress, and amino acid metabolism as well as metabolites that contribute to membrane integrity and neurotransmitters were significantly altered by HSD exposure. The results obtained from the NMR-based metabolomics approach together with biochemical assessment strongly suggest that hyperbaric decompression in a heliox saturation environment induces changes in several metabolites and that oxidative stress, energy

**Table 2. Summary of a panel of discriminant metabolites identified to be significantly changed between heliox saturation-hyperbaric-exposed and control rats in the three brain regions of cerebral cortex tissues, hippocampus, and striatum.**

| metabolites | FCᵃ (rᵇ, VIPᶜ, pᵈ) in cortex | FC (r, VIP, p) in hippocampus | FC (r, VIP, p) in striatum |
|---|---|---|---|
| ATP | / | 0.72(-0.72, 1.58, 0.03) | 0.62(-0.68, 1.38, 0.04) |
| Ala | 0.93(-0.54, 1.15, 0.13) | 0.89(-0.53, 1.4, 0.06) | / |
| AMP | 1.31(0.79, 1.59, 0.00) | 1.1(0.62, 1.34, 0.05) | / |
| Asc | / | 0.87(-0.53, 1.13, 0.04) | / |
| Asn | 0.85(-0.55, 1.12, 0.07) | / | / |
| Asp | / | / | 0.93(-0.66, 1.34, 0.06) |
| Car | 0.82(-0.59, 1.22, 0.02) | / | / |
| Cho | 0.64(-0.79, 1.66, 0.01) | 0.8(-0.62, 1.35, 0.1) | 0.63(-0.92, 1.86, 0.00) |
| Cyt | 0.8(-0.59, 1.26, 0.04) | 0.69(-0.73, 1.56, 0.01) | 0.61(-0.74, 1.49, 0.00) |
| DMA | / | 0.39(-0.56, 1.25, 0) | / |
| FMA | 1.26(0.51, 1.09, 0.08) | / | / |
| GABA | 0.89(-0.54, 1.18, 0.05) | 0.92(-0.82, 1.74, 0) | 0.86(-0.53, 1.07, 0.04) |
| Gln | / | / | 0.9(-0.5, 1.23, 0.06) |
| Gly | / | 0.89(-0.6, 1.4, 0.02) | / |
| GSH | 0.75(-0.75, 1.52, 0.01) | 0.82(-0.57, 1.21, 0.05) | / |
| Ile | / | / | 1.11(0.62, 1.33, 0.03) |
| IMP | / | 0.79(-0.72, 1.56, 0) | 0.82(-0.55, 1.13, 0.09) |
| Ino | 0.53(-0.92, 1.86, 0) | / | 0.68(-0.65, 1.3, 0.06) |
| Lac | 0.88(-0.57, 1.15, 0.02) | 0.87(-0.65, 1.6, 0.02) | / |
| Leu | / | / | 1.09(0.64, 1.35, 0.02) |
| Lys | / | / | 1.05(0.55, 1.27, 0.11) |
| NAA | / | / | 0.87(-0.51, 1.23, 0.09) |
| NAD+ | / | / | 0.84(-0.58, 1.25, 0.03) |
| NADP+ | / | / | 0.57(-0.67, 1.37, 0.02) |
| Nic | 1.33(0.67, 1.44, 0) | / | / |
| Pcho | 0.88(-0.48, 1.2, 0.07) | / | 0.87(-0.54, 1.1, 0.08) |
| Phe | 1.2(0.68, 1.48, 0.02) | / | / |
| Suc | / | 0.78(-0.5, 1.34, 0.02) | 0.83(-0.56, 1.28, 0.08) |
| Tau | / | 0.92(-0.39, 1.17, 0.21) | 0.79(-0.72, 1.5, 0.01) |
| Thr | / | 1.09(0.67, 1.57, 0.01) | / |
| Tyr | 0.78(-0.7, 1.43, 0.01) | 0.9(-0.5, 1.26, 0.18) | 0.74(-0.53, 1.19, 0.04) |
| Ura | 0.83(-0.43, 1.07, 0.21) | 0.37(-0.72, 1.55, 0.01) | 0.69(-0.52, 1.07, 0.07) |
| Uri | 0.68(-0.88, 1.8, 0) | 0.78(-0.58, 1.35, 0.11) | / |
| Val | / | / | 1.09(0.59, 1.35, 0.04) |

ᵃFold change (FC) between HSD-exposed rats and controls. ᵇ r with a positive value indicates a relatively higher concentration present in HSD-exposed rats, while negative values indicate a relatively lower concentration compared to the normal control. ᶜ VIP, Variable importance in projection. ᵈ p values indicate statistically significant changes between the two groups using a t test with a nonparametric test. The abbreviations of metabolites are shown in S1 Table. The value of correlation value r, VIP and *p* from Student's t test labeled in the brackets. The cutoff value of |r| is 0.50. The cutoff value of VIP was 1.0. The value of *p* defining statistically significant differences was less than 0.05. The metabolites meeting one of three criteria can be judged as discriminant metabolites.

metabolism and neurotransmitter alteration is a major mechanism for alterations in cerebral region-specific metabolomics profiles in HSD model rats. The data showed that multiple metabolic pathways, including glutathione metabolism, mitochondrial energy metabolism, glycolysis, BCAA metabolism, alanine, aspartate and glutamate metabolism, and neurotransmitter metabolism with brain region-specific metabolic disorders, were involved in the effects of 400msw-HSD.

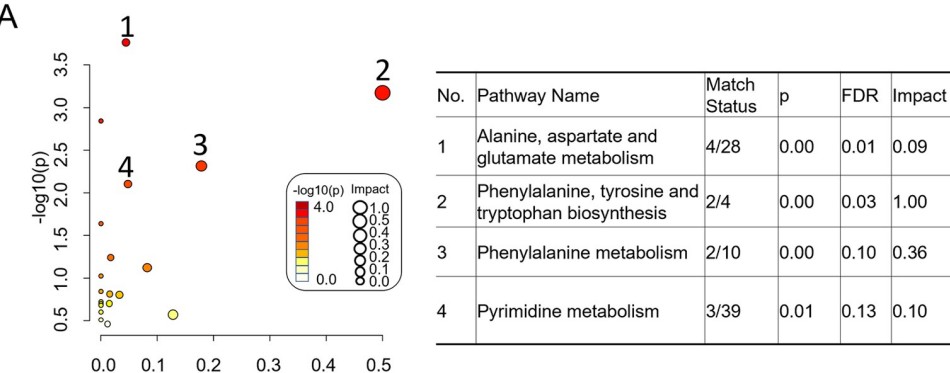

Quantitative pathway enrichment analysis of HSDC group compared with CONC group

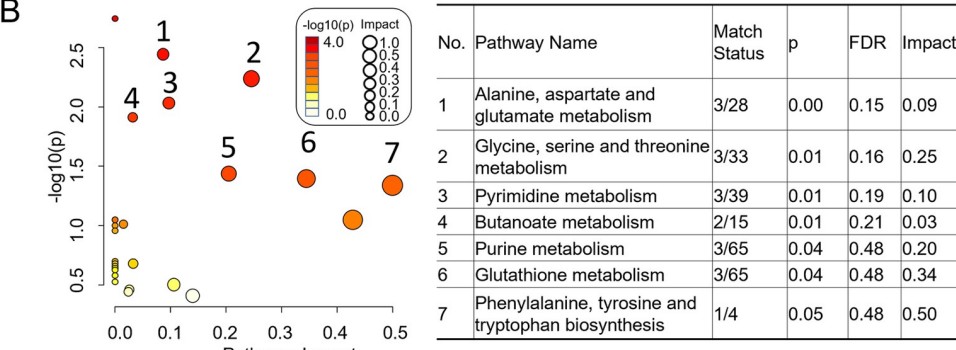

Quantitative pathway enrichment analysis of HSDH group compared with CONH group

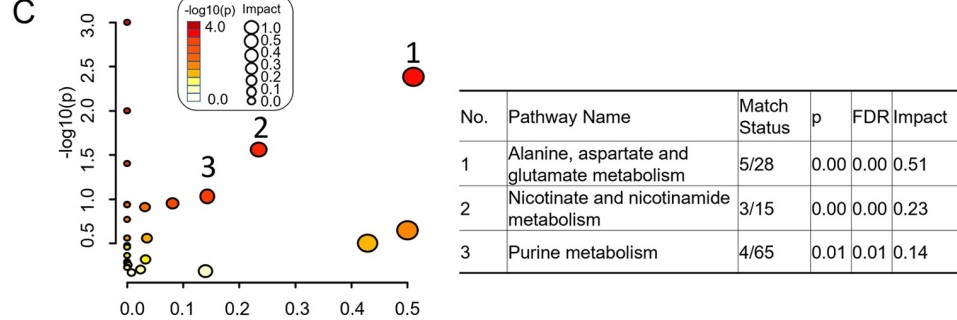

Quantitative pathway enrichment analysis of HSDS group compared with CONS group

**Fig 6. Quantitative pathway enrichment analysis.** Pathway topology plots illustrated by pathway impact values (which are listed in the right tables as well as FDR and p values) and–log10(p) from statistically significant metabolites from the cortex (A), hippocampus (B), and striatum (C) in the HSD groups relative to the controls.

## Oxidative stress analysis

During large depth (>100 msw) saturation diving, the central nervous system is continuously exposed to oxidative stress due to the excessive production of reactive oxygen species (ROS) triggered by the heliox-saturation pressurized environment [23, 24]. An imbalance between oxidant and antioxidant levels is a common metabolic regulatory element in various neurological disorders, including Alzheimer's disease [25], autism spectrum disorder [26], ischemic brain [21], traumatic brain injury [27], and so on [18, 28, 29]. The level changes of SOD, MDA, and Gpx, as recognized indicators for oxidative stress, reflect the oxygen free radical

injury in vivo. In the present study, the oxidative damage indicator (MDA) content was increased, and the antioxidant indicator (SOD and Gpx) activities were decreased in the model rats, suggesting that oxidative stress did occur in the cerebral cortex of the HSD rat model. Several other metabolites are also related to the regulation of oxidative stress. Taurine plays broader roles with antioxidant, anti-inflammatory, anti-apoptotic, osmolytic, and neuro-modulator effects to ameliorate the histopathological changes in brain and neuronal activity [30–36]. The significant decline in taurine content in the cerebral hippocampus and striatum upon HSD rats as well as another oxidative stress-related metabolite, GSH, was reduced in the cortex and hippocampus, suggesting that oxidative injury was induced in the different compartments of the HSD rat brain. GSH is widely recognized as an antioxidant quencher and produces stable molecules such as GSSH- and AKA-oxidized glutathione disulfide with the reaction of ROS. The low GSH level after HSD would affect mitochondrial function and redox balance, thereby accounting for the observed strong negative correlation between GSH *vs* Suc in the HSDH samples (Fig 5B' upper panel). Asc (aka Vitamin C) is considered an important antioxidant/micronutrient for its antioxidant capabilities and thus performs essential functions within brain neuronal maintenance. Herein, we report that Asc is at lower concentrations in HSDH samples than in controls. These findings are in agreement with each other, which suggests a direct link between oxidative stress and hyperbaric decompression in a heliox-saturated environment. Lower concentrations of antioxidant quenchers in the brain may be directly associated with decreased antioxidant capacity, thereby downregulating the generation of superoxides, including SOD and Gpx, and upregulating the levels of MDA.

## Energy metabolic pathway analysis

The statistically significant positive correlation between Ala and Lac in the HSDC and CONC groups indicated that Ala, as an amino acid, has a significantly higher link to anaerobic glycolysis [37]. LDH catalyzed the conversion of pyruvate into Lac, and the decreased activities of LDH in the cortex downregulated the expression of Lac, indicating a decrease in anaerobic pathways. However, the concentration of Lac favored energetic metabolism through the activity of aspartate aminotransferase (AST), alanine aminotransferase (ALT), and LDH, which can also be incorporated into the glutamate, glutamine, and GABA cycles in neurons. In this study, compared with CON rats, a significantly higher concentration of FMA in the HSDX group and a decreased concentration of Suc may indicate an energy metabolism decline, which was further confirmed by an increased AMP and a decreased ATP produced mainly by glycolysis and the TCA cycle. Na-K-ATPase, a key enzyme for the maintenance of a proper electrochemical gradient of sodium ions across the cell membrane, requires approximately 50% of the energy available to the brain [38, 39]. The malfunction of Na-K-ATPase has an essential role in the development of neurodegenerative diseases [40, 41]. The decreased activity of Na-K-ATPase in the cerebral cortex of the HSD group provided additional proof for the decreased energy metabolism induced by the HSD effect. Together, energy metabolism dysfunctions induced by hyperbaric HSD injury might include the collaboratively suppressed aerobic metabolism and anaerobic metabolism, which is a rare metabolic alteration phenomenon.

The levels of BCAAs (Ile, Val, and Leu) in the HSDS samples were upregulated in comparison with those in CONS samples. BCAAs can be converted into acetyl-CoA and succinyl-CoA [42] as substrates for gluconeogenesis and ATP generation. Thus, the upregulation of BCAAs was potentially induced to fulfill the energy compensation needs after HSD injury. Meanwhile, consistent with the energy compensation demand, the fatty acid β-oxidation metabolic pathway was also modulated, and a lower level of Car (a marker metabolite for fatty acid β-oxidation) in the HSDC samples was detected. The glycine, serine and threonine metabolism

pathway (Fig 6B) also supplies important energy metabolism precursors to enter the citrate cycle [43]. In this pathway, Cho, Gly, and Thr are the three hits. Cho and Gly were shown to decrease with HSD exposure. Likewise, cholinergic pathways have been linked to social and behavioral abnormalities, also as the essential component of cellular membranes and necessary for the synthesis of the neurotransmitter acetylcholine. Gly is the simplest amino acid with several functions, including fat metabolism, neurological function, muscle development, and incorporation into the antioxidant glutathione [44].

## Neurotransmitter metabolism

Neurotransmitter metabolism is vital to maintain normal brain function. However, a wide range of neurotransmitter imbalances has been characterized in HSD model rats. Increased levels of the excitatory transmitters dopamine and noradrenaline were accompanied by decreased inhibitory neurotransmitters including 5HT, Gly, and GABA in the HSD group compared with the controls. GABA, as the major inhibitory neurotransmitter, is responsible for halting excitatory glutamatergic activity, so naturally, disruption to either of these metabolites will affect the other in terms of the changes in the concentrations of Asp, Gly, and Gln. The excitatory neurotransmitter ACh is involved in multiple central nervous system functions [45] mainly by modulating acetylcholine receptors and their downstream pathways. In this study, AChE activity in the ACh hydrolysis process was measured. The decreased AChE activity indicates disturbance of the cholinergic pathway. Moreover, the striatal level of the essential amino acid Lys was found to be significantly elevated in HSD rats. Lys has been reported to block serotonin receptors, and its accumulation will affect the normal function of 5HT [46]. Asp, another excitatory neurotransmitter, is directly derived by transamination from a TCA cycle intermediate, oxaloacetate. We found that the concentration of Asp was significantly decreased in the striatum of HSD rats, providing additional proof for the imbalance effect on excitability neurotoxicity, with three other hits of Ala, FMA, and GABA. These four metabolites consist of the pathways of alanine, aspartate and glutamate metabolism (Figs 6A, 7). These findings indicate a perturbation in neurotransmitter recycling/production and the imbalance between the excitatory/inhibitory neurotransmitters induced by the HSD effect.

Taken together, systematic metabolic disorders, including energy metabolism dysfunctions, oxidative stress, and neurotransmitter metabolism disturbances, were induced in the region-specific cerebral injury of HSD rats. Downregulation of NAA, which is a general indicator of neuronal health, suggested neurofunctional abnormalities. These data indicated that neuronal damage was induced by the 4.0 MPa hyperbaric decompression exposure. In agreement with this deduction, the levels of Pcho and Cho, which are precursors of phosphatidylcholine (PC) involved in cell membrane lysis, apoptosis, and inflammatory responses, were downregulated. The concentration decrease in membrane-relevant metabolites suggested the disruption of cell membrane integrity related to neuronal damage. Neuronal damage might be the primary reason for the metabolic alteration induced by HSD effects.

## Conclusions

In the present study, NMR-based metabolomics and biochemistry assessment were applied to profile the metabolic alterations in the brain region-specific metabolic alteration of HSD model rats following 400 msw hyperbaric decompression of heliox saturation exposition. We then found that the HSD significantly induced metabolic aberrations, including oxidative stress, energy metabolism disorder, neurotransmitter metabolism disturbance, and cell membrane disruption. However, the more profound molecular mechanisms of HSD exposure should be investigated in future research.

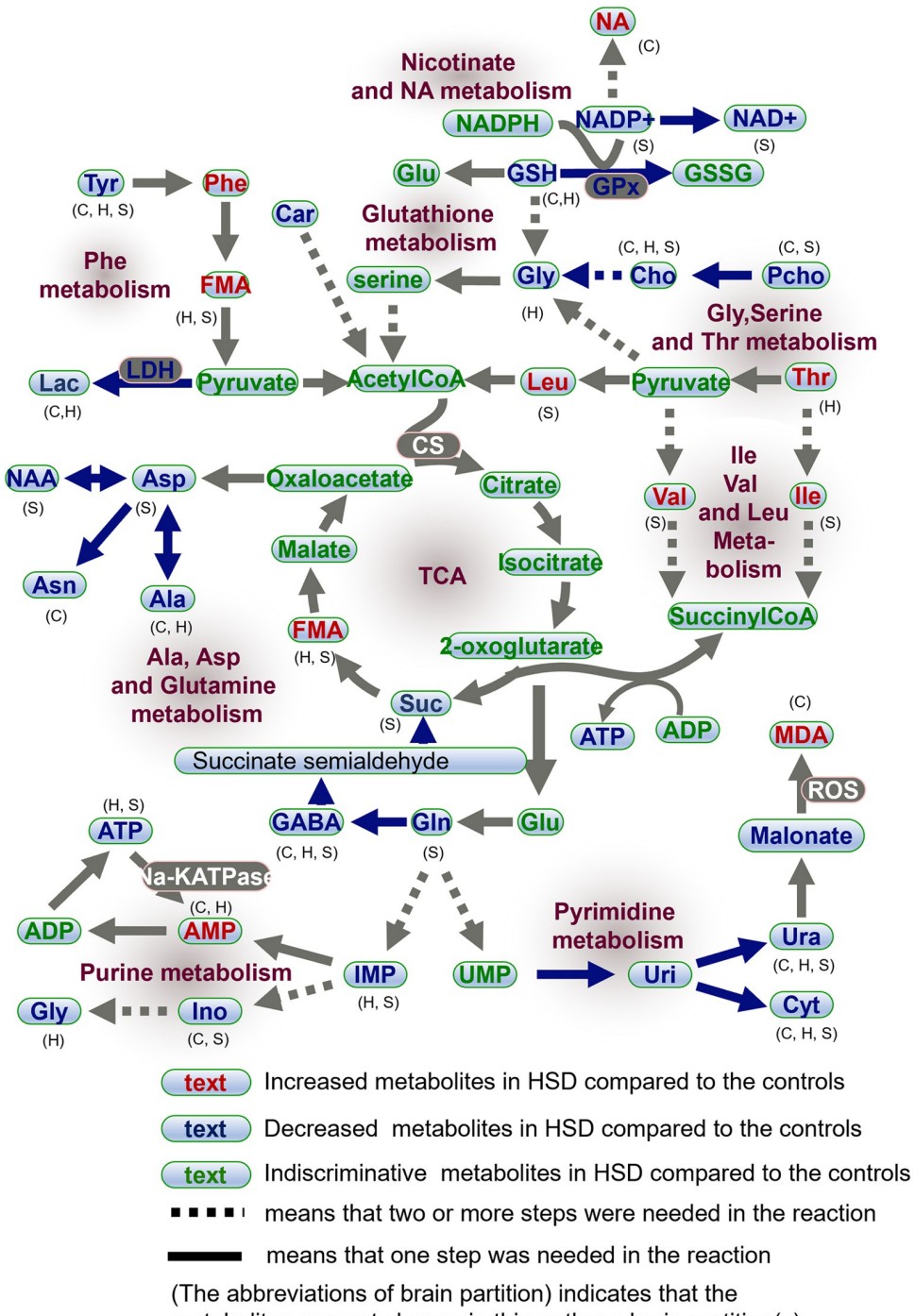

**Fig 7. Schematic overview of inferred changes in the potential cerebral metabolic pathways post hyperbaric heliox saturation.**

## Limitations

The potential sources of errors in this research might be the design and the analyses. Although cortex, hippocampus, and striatum dysfunctions have been strongly correlated with several

symptoms of CNS leisure in animal models, these three brain regions were the ones studied herein. The series of psychomotor and cognitive manifestations of CNS leisure under pathological factors (excessive atmospheric pressure, gas bubbles in the body, and decompression sickness) were also influenced by cerebellar dysfunction, but metabolic profile disturbance of the cerebellum was not considered in this research. Furthermore, only limited brain metabolites could be detected by NMR in this study; thus, an LC–MS (liquid chromatography tandem mass spectrometry)-based metabolomic approach should be used in future studies to detect more endogenous metabolites in the brain to fully understand the mechanism of the hyperbaric decompression effect in a heliox-saturated environment.

## Supporting information

**S1 Fig. Metabolic alterations driven by high pressure exposure of 400msw heliox saturation environment and across different brain regions.** Principal component analysis (PCA) scores plot of PC1/PC2 obtained from 1H NMR data and colored according to groups of CONC and HSDC (A, R2X = 0.40, Q2 = -0.02), CONH and HSDH (B, R2X = 0.50, Q2 = 0.16), and CONS and HSDS (C, R2X = 0.43, Q2 = 0.05),; scaling was done to unit variance; Partial least squares discriminant analysis (PLS-DA) scores plot from 1H NMR spectra of extracts from the cortex (A', R2X = 0.38, Q2 = 0.54), the hippocampus (B', R2X = 0.37, Q2 = 0.35), and striatum (C', R2X = 0.39, Q2 = 0.42) from CON and HSD groups; scaling was done to unit variance.
(DOCX)

**S1 Table. NMR Resonance assignments of 47 aqueous metabolites in $^1$H NMR spectra of brain samples.**
(DOCX)

**S2 Table. The buckets mean of peak height from discriminative aqueous metabolites in 1H NMR spectra acquired on the cerebral, hippocampus, striatum samples of HSD rats comparted to that of CON rats.**
(DOCX)

**S3 Table. The limit of detection of assay kits used in the present work.**
(DOCX)

**S4 Table. The peak height raw data of cortex samples.**
(DOCX)

**S5 Table. The peak height raw data of Hippocampus samples.**
(DOCX)

**S6 Table. The peak height raw data of striatum samples.**
(DOCX)

## Acknowledgments

The NMR data were recorded in the institutional technological service center of Shanghai Institute of Materia Medica, Chinese Academy of Sciences.

## Author Contributions

**Conceptualization:** Xia Liu, Yiqun Fang, Ci Li.

**Data curation:** Xia Liu, Tao Yang, Ji Xu.

**Formal analysis:** Xia Liu.

**Funding acquisition:** Ci Li.

**Investigation:** Xia Liu, Jia He, Wenwu Liu, Xuhua Yu, Yukun Wen, Ci Li.

**Methodology:** Xia Liu, Ji Xu, Jia He.

**Project administration:** Xia Liu, Ci Li.

**Resources:** Xia Liu.

**Supervision:** Xia Liu, Tao Yang, Ji Xu, Wenwu Liu, Xuhua Yu, Naixia Zhang, Ci Li.

**Validation:** Xia Liu, Jiajun Xu, Yukun Wen, Ci Li.

**Visualization:** Xia Liu, Jia He.

**Writing – original draft:** Xia Liu.

**Writing – review & editing:** Xia Liu, Yiqun Fang, Ci Li.

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
