## [Decision Letter · Decision Letter 0]

18 Nov 2022

PONE-D-22-24369Oxidative stress, dysfunctional energy metabolism, and destabilizing neurotransmitters altered cerebral metabolic profile in a rat model of a simulated heliox saturation diving to 4.0 MpaPLOS ONE

Dear Dr. Li,

Thank you for submitting your manuscript to PLOS ONE. After careful consideration, we feel that it has merit but does not fully meet PLOS ONE’s publication criteria as it currently stands. Therefore, we invite you to submit a revised version of the manuscript that addresses the points raised during the review process.

We look forward to receiving your revised manuscript.

Kind regards,

Ch Ratnasekhar, Ph.D.

Academic Editor

PLOS ONE

Journal Requirements:

2. In the Methods section of your revised manuscript, please include the full name of the institutional review board or ethics committee that approved the protocol, the approval or permit number that was issued, and the date that approval was granted.

“No”

7. Your ethics statement should only appear in the Methods section of your manuscript. If your ethics statement is written in any section besides the Methods, please move it to the Methods section and delete it from any other section. Please ensure that your ethics statement is included in your manuscript, as the ethics statement entered into the online submission form will not be published alongside your manuscript.

Reviewers' comments:

Reviewer's Responses to Questions

**Comments to the Author**

1. Is the manuscript technically sound, and do the data support the conclusions?

Reviewer #1: Partly

Reviewer #2: Yes

2. Has the statistical analysis been performed appropriately and rigorously? 

Reviewer #1: Yes

Reviewer #2: Yes

3. Have the authors made all data underlying the findings in their manuscript fully available?

Reviewer #1: Yes

Reviewer #2: Yes

4. Is the manuscript presented in an intelligible fashion and written in standard English?

Reviewer #1: No

Reviewer #2: Yes

5. Review Comments to the Author

Reviewer #1: The findings are interesting; however, the conclusion of manuscript should be evaluated as the changes in the neurotransmitter and other prooxidant and antioxidant parameters shown were transient or irreversible in nature. Please elaborate?

Is there any effect of pentobarbital sodium perse on the levels of neurotransmitter?

Please be consistent with the style of writing like “Five-hundred microliters” or 550 μL

Please abbreviate different labelling for the groups in the beginning “CONS”, “CONH”, “CONC”; “HSDS”, HSDC”, “HSDH”

Figure 3 legend labelling “cerebral” please correct

Please mention the limit of detection of the ELISA kits used in the present study

Table 1- Please correct “nmo” and “nmol”

Table 1 – For epinephrine value the standard deviation between CON and HSD is quite high to have a significant value? Please check.

What does the dotted and straight lines indicate in figure 7? Please mention.

Figure 6C – Table is not spaced properly to analyze the data.

Figure 6C – Please mention the use of color and what it denotes?

There is no mention of metabolic enzymes, neurotransmitter levels and oxidative stress indicators in hippocampus and striatum. Please explain?

Reviewer #2: The research article “Oxidative stress, dysfunctional energy metabolism, and destabilizing neurotransmitters altered cerebral metabolic profile in a rat model of a simulated heliox saturation diving to 4.0 Mpa” by Liu et al is very interesting and is expected to add significant new information in the research area of harmful effects of deep sea diving. However, I have some minor concerns mentioned below:

1. Could the metabolomic and biochemical alterations in rat brain induced by HSD be reversible? Has the author tried to assess these parameters after giving rats some recovery time?

2. Are the observed changes dependent on the time for which diving is performed?

6. PLOS authors have the option to publish the peer review history of their article (what does this mean?). If published, this will include your full peer review and any attached files.

Reviewer #1: No

Reviewer #2: No

---

## [Author Response · Author response to Decision Letter 0]

15 Feb 2023

rebuttal letter

Dear Reviewers and editors, 

Thank you for giving us a chance to submit a revised version of the manuscript.

Journal Requirements:

Answer: We appreciate this comment. Changes have been made according to the manuscript format.

2. In the Methods section of your revised manuscript, please include the full name of the institutional review board or ethics committee that approved the protocol, the approval or permit number that was issued, and the date that approval was granted.

Answer: We appreciate this comment. Changes have been made as follow: All experimental protocols were approved by the Animal Ethics Committee of Naval Medical Center of PLA, Naval Medical University (Approval no: SYXK(Shanghai)2017-0019, Approval year: 2020).

“No”

d) If you did not receive any funding for this study, please state: “The authors received no specific funding for this work.”235-247-341

Answer: We appreciate this comment. This work was financially supported by the military medical innovation special project (CWS11J028) and the Thirteenth Five-Year Plan" military key disciplines and professional construction projects (Grant No. 2020SZ19-4; 22020SZ20-3; 2020SZ23-1; 2020SZ25). The funders had no role in study design, data collection and analysis, decision to publish, or preparation of the manuscript. Not any authors received a salary from any of above-mentioned funders.

Answer: We appreciate this comment. Metabolomics data have been deposited to the EMBL-EBI MetaboLights database (DOI: 10.1093/nar/gkz1019, PMID:31691833) with the identifier MTBLS7185. The complete dataset can be accessed here https://www.ebi.ac.uk/metabolights/MTBLS7185.

Answer: We appreciate this comment. The ORCID iD of corresponding author (Ci Li) is 0000-0002-9572-1155. 

Answer: We appreciate this comment. We deleted these sentences of” In addition, we also investigated the serum metabolic profile changes in this research. Increased levels of isobutyrate and trimethylamine and decreased levels of lactate and pyruvate were present in the serum samples of the HSD group compared with those of the CON group (data not shown). Such a small number of differential metabolites in serum suggest that the brain tissue (34 discriminative metabolites) might be the target organ of hyperbaric exposure. Certainly, additional data are needed to draw more weighty conclusions.”

7. Your ethics statement should only appear in the Methods section of your manuscript. If your ethics statement is written in any section besides the Methods, please move it to the Methods section and delete it from any other section. Please ensure that your ethics statement is included in your manuscript, as the ethics statement entered into the online submission form will not be published alongside your manuscript.

Answer: We appreciate this comment. We move the “Ethics approval” to the methods.

Answer: We appreciate this comment. We added “Additional file: Additional file 1: Supplementary materials. “in the end of manuscript.

Answer: We appreciate this comment. We reviewed the reference list. We didn’t cite any paper that have been retracted.

Reviewers' comments:

Reviewer's Responses to Questions

Comments to the Author

1. Is the manuscript technically sound, and do the data support the conclusions?

Reviewer #1: Partly

Reviewer #2: Yes

2. Has the statistical analysis been performed appropriately and rigorously?

Reviewer #1: Yes

Reviewer #2: Yes

3. Have the authors made all data underlying the findings in their manuscript fully available?

Reviewer #1: Yes

Reviewer #2: Yes

4. Is the manuscript presented in an intelligible fashion and written in standard English?

Reviewer #1: No

Reviewer #2: Yes

5. Review Comments to the Author

Reviewer #1: The findings are interesting; however, the conclusion of manuscript should be evaluated as the changes in the neurotransmitter and other prooxidant and antioxidant parameters shown were transient or irreversible in nature. Please elaborate?

Answer: We appreciate this comment. Whether the changes in the neurotransmitter and other prooxidant and antioxidant parameters shown were transient and irreversible in nature in the HSD group, we will elaborate in a future paper.

Is there any effect of pentobarbital sodium perse on the levels of neurotransmitter?

Answer: We appreciate this comment. We searched the literatures and found that Pentobarbital antagonizes the A1 adenosine receptor-mediated inhibition of hippocampal neurotransmitter release. Thus, there is some effect of pentobarbital sodium perse on the levels of neurotransmitter. But all the rats received pentobarbital sodium dose proportional to theirs body weight, the effect might also proportionate to body weight, the body weight between HSD and CON groups showed no statistically significant meaning. Thus，We can infer that the statistical pattern of neurotransmitter changes is not affected by pentobarbital sodium. Of course, we will confirm this statement in subsequent experimental studies where animals are not given sodium pentobarbital.

Please be consistent with the style of writing like “Five-hundred microliters” or 550 μL. 

Answer: We appreciate this comment. “Five-hundred microliters” was changed as “an aliquot of 500 μL”.

Please abbreviate different labelling for the groups in the beginning “CONS”, “CONH”, “CONC”; “HSDS”, HSDC”, “HSDH”

Answer: We appreciate this comment. We changed the sentence” Cortex, hippocampus, and striatum tissues were dissected” as “Cortex, hippocampus, and striatum tissues were dissected, and abbreviated respectively as HSDC, HSDH, HSDS, CONC, CONH and CONS in two animal groups”.

Figure 3 legend labelling “cerebral” please correct

Answer: We appreciate this comment. we changed the cerebral as “cortex”.

Please mention the limit of detection of the ELISA kits used in the present study

Answer: We appreciate this comment. We listed the information of assay kits in the supply materials.

Company Product name Limit of detection 

Abcam ab83355 ATP Assay Kit 1 µM

Abcam ab235937 Cholinesterase Activity Assay Kit 1 mU/ml

Abcam ab102526 Lactate Dehydrogenase Assay Kit 1 mU/ml

RD Universal Dopamine ELISA Kit 18.75 pg/mL

Abnova KA3768 Epinephrine/Norepinephrine ELISA Kit Adrenaline: 0.25 ng/mL 

Abnova KA3768 Epinephrine/Norepinephrine ELISA Kit Noradrenaline: 0.1 ng/mL 

BioSource Rat 5-Hydroxytryptamine Assay Kit 1.0 ng/mL

Santa Cruz Rat Gamma-aminobutyric acid (GABA) Assay Kit 0.1μmol/mL

cayman Rat Super Oxidase Dimutase (SOD) ELISA Kit 1.0 U/mL

cayman Rat malondialchehyche (MDA) ELISA Kit 0.1 nmol/mL

cayman Rat Glutathione peroxidase (GSH-Px) ELISA Kit 1.0 U/mL

Table 1- Please correct “nmo” and “nmol”

Answer: We appreciate this comment. We corrected “nmo” and “nmol”.

Table 1 – For epinephrine value the standard deviation between CON and HSD is quite high to have a significant value? Please check

Answer: We appreciate this comment. A decimal point has been omitted and “11.42±119” of HSD was changed as “11.42±1.19”. we are very sorry for this mistake.

What does the dotted and straight lines indicate in figure 7? Please mention.

Answer: We appreciate this comment. The dotted lines in the figure 7 mean that two or more Intermediate entities or steps were needed in the transformation process of the first metabolites to the second metabolites. The straight lines mean that metabolites A can converted to metabolites B in one step reaction.

Figure 6C – Table is not spaced properly to analyze the data.

Answer: We appreciate this comment and added lines inside the table, thus data can be analyzed properly. 

Figure 6C – Please mention the use of color and what it denotes?

Answer: We appreciate this comment, the shades of color in the circles in the Figure 6C stands for the value of -log10(p), the higher the redder, the lower the color become white from yellow.

There is no mention of metabolic enzymes, neurotransmitter levels and oxidative stress indicators in hippocampus and striatum. Please explain?

Answer: We appreciate this comment, for the volume and weight of hippocampus and striatum were so small, thus the separated tissues from rat’s brain were only for the detection of NMR-based metabolome.

Reviewer #2: The research article “Oxidative stress, dysfunctional energy metabolism, and destabilizing neurotransmitters altered cerebral metabolic profile in a rat model of a simulated heliox saturation diving to 4.0 Mpa” by Liu et al is very interesting and is expected to add significant new information in the research area of harmful effects of deep sea diving. However, I have some minor concerns mentioned below:

1. Could the metabolomic and biochemical alterations in rat brain induced by HSD be reversible? Has the author tried to assess these parameters after giving rats some recovery time?

Answer: We appreciate this comment. Whether the metabolomic and biochemical alterations in rat brain induced by HSD could be reversible, we will observe the changes of these parameters in the future by trying to give rats some recovery time.

2. Are the observed changes dependent on the time for which diving is performed?

Answer: We appreciate this comment. we will investigate whether the observation will change dependent on the diving time in the future.

6. PLOS authors have the option to publish the peer review history of their article (what does this mean?). If published, this will include your full peer review and any attached files.

Do you want your identity to be public for this peer review? For information about this choice, including consent withdrawal, please see our Privacy Policy.

Reviewer #1: No

Reviewer #2: No

---

## [Editor Report · Decision Letter 1]

22 Feb 2023

Oxidative stress, dysfunctional energy metabolism, and destabilizing neurotransmitters altered cerebral metabolic profile in a rat model of a simulated heliox saturation diving to 4.0 Mpa

PONE-D-22-24369R1

Dear Dr. Ci Li,

We’re pleased to inform you that your manuscript has been judged scientifically suitable for publication and will be formally accepted for publication once it meets all outstanding technical requirements.

Kind regards,

Ch Ratnasekhar, Ph.D.

Academic Editor

PLOS ONE
---

## [Editor Report · Acceptance letter]

6 Mar 2023

PONE-D-22-24369R1 

Oxidative stress, dysfunctional energy metabolism, and destabilizing neurotransmitters altered the cerebral metabolic profile in a rat model of simulated heliox saturation diving to 4.0 MPa 

Dear Dr. Li:

I'm pleased to inform you that your manuscript has been deemed suitable for publication in PLOS ONE. Congratulations! Your manuscript is now with our production department. 

Kind regards, 

on behalf of

Dr. Ch Ratnasekhar 

Academic Editor

PLOS ONE